# Community health worker knowledge, attitudes and practices towards COVID-19: Learnings from an online cross-sectional survey using a digital health platform, UpSCALE, in Mozambique

**Mitra Feldman[1], Vera Lacey Krylova[2], Poppy Farrow[2], Laura Donovan[2], Edson Zandamela[3], Joaquim Rebelo[3], Maria Rodrigues[3], Antonio Bulo[3], Carlos Ferraz[3], Humberto Rodrigues[4], Arantxa Roca-Feltrer[2], Kevin Baker** [2,5] *

1 Independent Consultant, Costa Rica, 2 Malaria Consortium, London, United Kingdom, 3 Malaria Consortium, Maputo, Mozambique, 4 Republic of Mozambique Ministry of Health, Maputo, Mozambique, 5 Department of Global Public Health, Karolinska Institute, Stockholm, Sweden

* k.baker@malariaconsortium.org

## Abstract

Healthcare workers (HCWs) are at the frontline of the Coronavirus Disease 2019 (COVID-19) pandemic response, yet there is a paucity of literature on their knowledge, attitudes and practices (KAP) in relation to the pandemic. Community Health Workers (CHWs) in Mozambique are known locally as agentes polivalentes elementares (APEs). While technical guidance surrounding COVID-19 is available to support APEs, communicating this information has been challenging due to restrictions on travel, face-to-face group meetings and training, imposed from May to August 2020. A digital health platform, upSCALE, that already supports 1,213 APEs and 299 supervisors across three provinces, is being used to support APEs on effective COVID-19 management by delivering COVID-19 sensitive SMS messages, training modules and a COVID-19 KAP survey. The KAP survey, conducted from June 2020 to August 2020, consisted of 10 questions. Of 1,065 active upSCALE APEs, 28% completed the survey. Results indicate that only a small proportion of APEs listed the correct COVID-19 symptoms, transmission routes and appropriate prevention measures (n = (25%), n = (16%) and n = (39%), respectively) specifically included in national health education materials. Misconceptions were mainly related to transmission routes, high risk individuals and asymptomatic patients. 84% said they followed all government prevention guidelines. The results from the KAP survey were used to support the rapid development and deployment of targeted COVID-19 awareness and education materials for the APEs. A follow-up KAP survey is planned for November 2020. Adapting the existing upSCALE platform enabled a better understanding, in real time, of the KAP of APEs around COVID-19 management. Subsequently, supporting delivery of tailored messages and education, vital for ensuring a successful COVID-19 response.

**Data Availability Statement:** Data are available from the UK Data Service (DOI: 10.5255/UKDA-SN-854605).

**Funding:** Malaria Consortium US. The funders had no role in study design, data collection and analysis, decision to publish, or preparation of the manuscript.

**Competing interests:** The authors have declared that no competing interests exist.

**Abbreviations:** APE, Agentes polivalentes elementares; CHWs, Community health workers; COVID-19, Coronavirus disease 2019; DHIS, District Health Information System; HCWs, Health care workers; iCCM, Integrated community case management; KAP, Knowledge, attitudes and practices; MoH, Ministry of Health; PPE, Personal protective equipment; SARS-COV-2, Severe acute respiratory syndrome coronavirus 2; SMS, Short message service; WHO, World Health Organization.

## Introduction

Coronavirus Disease 2019 (COVID-19) is a rapidly expanding pandemic caused by a novel human coronavirus: severe acute respiratory syndrome coronavirus 2 (SARS-COV-2) [1]. It was first reported in December 2019 among patients with viral pneumonia symptoms in Wuhan, China [2]. As of 12 October 2020, there have been 37,423,660 confirmed cases of COVID-19, including 1,074,817 deaths, reported to WHO [3]. To date, the African continent appears to be one of the least affected regions in the world with 1,594,750 cases and 38,570 deaths [4], but the numbers are increasing [5]. Mozambique reported its first imported COVID-19 case on March 22, 2020 and has seen an increase in cases since then [6]. As of 12 October there have been a total of 10,001 confirmed cases and 71 related deaths in the country [3].

COVID-19 is transmitted from person-to-person through inhalation of aerosols from an infected individual [7]. Older adults and patients with pre-existing illnesses (like hypertension, cardiac disease, lung disease, cancer, or diabetes) have been identified as potential risk factors for severe disease and mortality [8]. To date, there is no antiviral curative treatment or vaccine that has been recommended for COVID-19 [9]. More information about its distribution, transmission, pathophysiology, treatment, and prevention are being studied. Infection by SARS-CoV-2 in humans occurs mainly through close contact through respiratory droplets, by direct contact with infected persons, or by contact with contaminated objects and surfaces [10]. Primary preventive measures include frequent hand washing for a minimum of 20 seconds at a time, maintaining social distance, and respiratory hygiene (covering mouth and nose while coughing or sneezing) [11].

Healthcare workers (HCWs) are at the frontline of the COVID-19 pandemic response [12], yet there is a paucity of literature on their knowledge, attitudes and practices (KAP) in relation to the COVID-19 pandemic. One HCW KAP study, focused on Asian HCWs and medical students, revealed that a significant proportion (61%) had poor knowledge of its transmission [13]. A study in Uganda, among hospital based HCWs, found that only 69% had sufficient knowledge of the disease, although a higher %age (74%) reported practicing good prevention measures [14]. One HCW study in Pakistan showed higher knowledge (93%) and good practice (89%) regarding COVID-19 [15]. Findings from another HCW KAP study in Nepal found 82% of respondents had good to moderate knowledge of COVID-19 and 84% had good prevention practices [16]. In a study among HCWs in China, 89% of HCWs had sufficient knowledge of COVID-19 and 90% followed correct practices regarding COVID-19 prevention, yet more than 85% feared self-infection with the virus [17].

Other KAP studies, in Nigeria and Tanzania, have focused on the KAP of COVID-19 among community residents. Residents from a study in Northern Nigeria showed high level of awareness of COVID-19 and prevention methods (99% and 95%, respectively). However, a lower %age (80%) said they follow government guidelines for prevention [18]. In the Tanzanian study, 84% of the participants had a good knowledge of COVID-19 [19]. Similar community based KAP studies have been conducted in China, the Philippines and Malaysia. A Chinese residents' KAP survey towards COVID-19, conducted during the rapid rise period of the outbreak, showed that COVID-19 knowledge was 90% and that nearly all of the participants (98%) wore masks when going out [20]. In a survey among poor households in the Philippines, 94% of respondents were aware of COVID-19 and 82% were aware of appropriate preventive measures to protect people from infection [21]. The Malaysia survey indicated that 81% had good knowledge of COVID-19, however, there was noticeable confusion among participants regarding transmission of the virus. Only 43% of participants answered correctly when asked if the virus was airborne and just 36% answered correctly when asked if eating and

touching wild animals could result in infection [22]. To our knowledge, there have been no published studies from sub-Saharan Africa to assess KAP toward COVID-19 specifically among community health workers (CHWs).

In Mozambique, CHWs, known locally as agentes polivalentes elementares (APEs), are typically community members with a basic level of education, usually up to grade 6 or 7, who are trained to provide basic healthcare services and conduct health promotion activities in the remote areas in which they live. This includes one month's training on the providing integrated community case management (iCCM) for malaria, pneumonia and diarrhoea for children aged 2–59 months and, as of 2014, family planning, pregnancy tracking, antenatal and post-partum care. While technical guidance surrounding COVID-19 surveillance, case definitions and testing strategies are available to support APEs, communicating this information to the wider APE network has been a significant challenge. These have been largely due to restrictions on travel and face-to-face group meetings and training, which were in place from the beginning of April through to the beginning of August, when routine face-to-face activities were resumed. Other challenges include incorporating COVID-19 surveillance into existing surveillance networks and diagnosing COVID-19 within the community, due to the similarity of symptoms with other routinely presented diseases such as malaria and pneumonia. Furthermore, due to the novelty of COVID-19, the information available globally is constantly evolving and expanding, and this in turn requires continuous reflection and analysis in terms of the impact on country strategies and plans. This, combined with limited data on community-level caseload due to a lack of accessible and affordable tests, poses significant challenges in the APEs' abilities to conduct their routine activities and support the COVID-19 response. There is need for a dynamic, rapid response model to effectively support APEs on COVID-19 management.

The upSCALE project, supported by UK AID (via UNICEF), is a continuation of the Bill and Melinda Gates Foundation supported inSCALE project. It consists of a smartphone app that guides APEs through patient registration, assists with diagnosis and advises on treatment and referrals (primarily related to iCCM), and a tablet-based app that allows supervisors to monitor CHW performance and stock levels. The applications were developed on an open source platform designed specifically for use by frontline health workers (CommCare, Dimagi). Data entered by APEs through the upSCALE app is visualised in the District Health Information System (DHIS2) at district, provincial and national levels of Mozambique's health system. The app has the potential to analyse local disease-specific trends in near real-time, allowing managers to improve their resource allocation. The programme is currently being implemented in three provinces (first in Inhambane in the south of Mozambique, followed by Cabo Delgado in the north, and then by Zambézia in central Mozambique) with 1,213 APEs and 299 supervisors using the app. The MoH is planning to roll out the platform to all 8,800 APEs nationally by 2021. APEs received 1 week of initial training on how to use upSCALE.

Given the lack of sufficient accurate and up-to-date information on COVID-19, especially in rural communities, there is a risk of high levels of disinformation, and possible questionable practices. At the same time, the most up-to-date advice about COVID-19 symptoms (which is constantly changing as we learn more about the virus) and how to protect community members from getting infected and infecting others might not reach all community members timely and/or effectively. The use of the upSCALE app presents a dynamic solution for COVID-19 messaging and tracking, capable of adapting and expanding messages as new info/ practices arise, also allowing to capture how communities are adapting to living in a pandemic such as COVID-19. In this study, we investigate the KAP of APEs who use a digital health platform UpSCALE, in three provinces in Mozambique. This is the first report on the knowledge, attitude, and practices of these community health workers from Mozambique. Findings from

this study should contribute to the global and local efforts to better control the COVID-19 pandemic.

## Methods

To support the COVID-19 response in Mozambique, Malaria Consortium, in partnership with Ministry of Health and Dimagi, our digital development partner, further developed the upSCALE platform and the use of telemedicine—through the use of short message service (SMS), training modules, monitoring of key indicators on routine services for women and children, stock control and disease surveillance.

To better support tailoring and targeting of appropriate messages a COVID-19 KAP survey was developed and delivered to the APEs via upSCALE from 9 June 2020 to 14 August 2020. The KAP was a cross-sectional study that was sent to all 1,400 upSCALE registered APEs and involved those who responded to an invitation via SMS message, asking them to complete the questionnaire on the UPSCALE application. The SMS messages were accompanied by simple videos demonstrating how to install and complete the KAP survey on the application.

The KAP survey questionnaire (see S1 Annex) was developed using various references including the WHO survey tools, guidance on COVID-19 insights and previous KAP surveys conducted during the pandemic [20, 23, 24]. The self-reported survey consisted of 9 questions related to COVID-19 symptoms, prevention and transmission, as well as main sources of information. The components of the knowledge section included the causes and modes of COVID-19 transmission, main symptoms, transmissibility from asymptomatic patients, individuals at risk and preventive measures. The attitudes and practices sections were comprised of questions related to COVID-19 preventive measures practiced, adherence to government prevention measures, barriers to following recommended measures and what to do when symptoms occur. The draft survey was pretested with four users before launch to ensure it was accessible and respondent friendly.

### Ethical considerations

The KAP survey was conducted as a programmatic activity under the upSCALE project, which is implemented in collaboration with the Mozambique MoH, therefore separate ethics approval was not sought. The questionnaire contained a consent section that included a statement about its purpose, objectives, voluntary participation, and a declaration of confidentiality and anonymity. All responses were submitted anonymously via the upSCALE platform.

## Results

Of the 1,456 APEs registered with the upSCALE app, 1,065 had operational phones at the time of the KAP survey and 297 of these completed the questionnaire (28%) over a three month period. Of these, 48% were from Cabo Delgado, 11% from Inhambane, and 41% from Zambézia. It took the respondents an average of five minutes to complete the survey. Female respondents made up 24% of those who completed the questionnaire (Table 1).

The largest group of respondents said they had heard of COVID-19 through the local radio (27%), which was followed closely by their health facility (26%). Approximately 15% said they had heard of COVID-19 through the upSCALE app. A similar proportion (13%) had heard via word of mouth. Newspapers (nine %) and television (five %) were also mentioned. A very small %age mentioned government websites (two %) and social media (1%). A few respondents (1%) said they had not heard of COVID-19 through any source (Table 2).

When looking at combined data from June–August 2020, 25% of respondents listed the correct three main symptoms of COVID-19, and no others (fever, cough and shortness of

**Table 1. Demographic characteristics of respondents (N = 297).**

| Characteristic | | Number | Percentage (%) |
|---|---|---|---|
| Gender | Male | 226 | 76 |
| | Female | 71 | 24 |
| Age | 18–29 | 83 | 28 |
| | 30–49 | 97 | 33 |
| | Above 50 | 117 | 39 |
| Province | Cabo Delgado | 143 | 48 |
| | Inhambane | 33 | 11 |
| | Zambezia | 121 | 41 |
| Years as an APEs | 0–2 years | 74 | 25 |
| | 3–5 years | 109 | 37 |
| | More than 5 years | 114 | 38 |

breath), as outlined in the Ministry of Health (MoH) guidelines [25]. When asked to list all symptoms of COVID-19; headache was the most frequently listed symptom (98%), followed by fever and dry cough (both listed by 90% of respondents). Shortness of breath was also mentioned by 87% of respondents (Table 3).

Regarding transmission routes, 16% listed the correct combination, according to MoH guidelines (direct contact with contaminated surfaces, respiratory droplets and direct contact with an infected person). When asked to list all possible transmission routes, direct contact with contaminated surfaces and objects was mentioned the most (91%), this was followed closely by direct contact with an infected person (90%) and respiratory droplets (89%). A large %age (86%) thought the virus is airborne, and 83% mentioned the emptying of latrines as a source of transmission. When presented with the statement "A person with COVID-19 without any symptoms cannot spread the virus,"41% said it was true (Table 3).

APEs were asked to list methods of COVID-19 prevention, with 39% listing the correct combination of preventions methods (wash hands, wear a mask, avoid touching your face, cover your mouth when you cough or sneeze, stay home if you feel unwell and practice social distancing of 1.5 meters). The most frequently listed method of prevention (when asked to list all) was covering your face while sneezing (97%). This was followed closely by staying home if you feel unwell (93%), avoiding touching your face, handwashing and wearing a mask (all mentioned by 92% of respondents), and maintaining a physical distance of 1.5 meters (91%) (Table 3).

**Table 2. Participants sources of knowledge of COVID-19 (N = 297).**

| Question | Option | Responses n (%) |
|---|---|---|
| 1. Where do you hear or see the messages about COVID-19? (select all that apply) | Newspapers | 27 (9) |
| | Word-of-mouth | 39 (13) |
| | Government website | 6 (2) |
| | Local television | 15 (5) |
| | Local radio | 80 (27) |
| | Through UpSCALE | 45 (15) |
| | Health facility | 77 (26) |
| | Other (please specify) | 3 (1) |
| | I have not heard any messages about COVID-19 | 3 (1) |

**Table 3. General knowledge of COVID-19 among respondent APES (N = 297).**

| Question | Options | Responses n (%) |
|---|---|---|
| What are the three main clinical symptoms of COVID-19? (Select three) | **Fever** | **267 (90)** |
| | Headache | 291 (98) |
| | **Shortness of breath** | **258 (87)** |
| | **Dry, persistent cough** | **267 (90)** |
| | Conjunctivitis | 133 (45) |
| | Fatigue | 149 (50) |
| | Diarrhoea | 163 (55) |
| | Loss of speech or movement | 30 (10) |
| **Correct Answer** | **All three correctly selected** | **74 (25)** |
| Which of the below are at risk groups for COVID-19? (Select all that apply) | **Elderly individuals (aged >70)** | **288 (97)** |
| | Pregnant women | 9 (3) |
| | **Those with chronic illnesses (e.g. heart disease, diabetes)** | **291 (98)** |
| | Children | 119 (40) |
| | **Obese individuals** | **291 (98)** |
| **Correct Answer** | **Three selected** | **6 (2)** |
| Which of the following are methods of preventing infection with COVID-19? (Select all that apply) | **Washing hands regularly with soap and water, or cleaning them with alcohol-based hand rub** | 273 (92) |
| | **Wearing a facemask** | 273 (92) |
| | **Avoid touching your face** | 273 (92) |
| | **Cover your mouth and nose when coughing or sneezing** | 288 (97) |
| | **Stay home if you feel unwell** | 276 (93) |
| | **Practice physical distancing by avoiding unnecessary travel, staying away from large groups of people and keeping 1.5m apart from others** | 270 (91) |
| **Correct Answer** | **All options correctly selected** | **116 (39)** |
| How is COVID-19 transmitted? (Select all that apply) | Through the air (airborne) | 255 (86) |
| | **Contact with contaminated objects and surfaces** | **270 (91)** |
| | **Respiratory droplets** | **264 (89)** |
| | **Direct contact with infected persons** | **267 (90)** |
| | Emptying latrines and handling of waste | 247 (83) |
| **Correct Answer** | **All three correctly selected** | **48 (16)** |
| What is the minimum length of handwashing time recommended to effectively prevent onward transmission of COVID-19? | 10 seconds | 62 (21) |
| | **20 seconds** | **48 (16)** |
| | 30 seconds | 45 (15) |
| | 60 seconds | 134 (45) |
| **Correct answer** | | **48 (16)** |
| A person infected with COVID-19 who does not show symptoms cannot spread the coronavirus. | True | 122 (41) |
| | **False** | **169 (57)** |
| | Unsure | 6 (2) |

*(Continued)*

**Table 3.** (Continued)

| Question | Options | Responses n (%) |
|---|---|---|
| **Correct answer** | | **169 (57)** |
| If you have symptoms of COVID-19, what measures should be taken? (Select all that apply) | **Self-isolate by staying at home for at least 7 days** | **244 (82)** |
| | **Get plenty of rest** | **83 (28)** |
| | **Stay hydrated and take paracetamol** | **71 (24)** |
| | **Contact your local health facility or Alô Vida via telephone** | **142 (48)** |
| | **Wear a facemask** | **154 (52)** |
| | **Monitor your symptoms regularly** | **89 (30)** |
| **Correct answer** | **All of the above** | **59 (20)** |

Almost half of the APEs (45%) said that the minimum recommended handwashing length was one minute, followed by 10 seconds (21%), 20 seconds (16%), and 30 seconds (15%) (Table 3).

When asked who was at the greatest risk, less than 2% (1.24%) of respondents correctly identified elderly individuals, obese individuals and those with an underlying chronic illness. Those with chronic illness and obese individual were listed the most frequently (each mentioned by 98%), followed by elderly individuals (97%), pregnant women (3%) and children (40%) (Table 3).

The next survey question asked participants to list all actions someone should take when COVID-19 symptoms develop, 20% of respondents provided the right combination of answers, and according to government guidelines (self-isolate, get plenty of rest, stay hydrated and take paracetamol, contact your local health facility, wear a mask, and monitor symptoms regularly).

Overall, self-isolate was included the most (82%), followed by wear a face mask (52%), contact your local health facility (48%), monitor symptoms regularly (30%), get plenty of rest (28%), and stay hydrated and take paracetamol (24%) (Table 3).

The majority of respondents (84%) said they followed all MoH prevention guidelines at all times, while 13% said they followed only some of the prevention guidelines (with wearing a mask being the most frequently cited) (Table 4). When asked "What is preventing you from fully protecting yourself" personal protective equipment (PPE) was mentioned by four %. However, lack of PPE decreased over time, with seven % mentioning it PPE in June, five % in July and only two % in August.

Another output from the KAP survey has been the development and deployment of seven COVID-19 sensitive health education modules, integrated onto upSCALE, including; hand

**Table 4. General attitudes and practices towards COVID-19 among respondent APEs (N = 297).**

| Question | Options | Responses n (%) |
|---|---|---|
| Do you adhere to the prevention measures set out by national health authorities? (E.g. regular handwashing, social distancing) | Yes–all of them | 249 (84) |
| | No | 0 |
| | Some of them or sometimes | 36 (12) |
| | Don't know | 12 (4) |

and respiratory hygiene, safe use and disposal of PPE and waste management. All materials are supported with multi-media tools, such as training and deployment videos, to improve the technique and practice of using the COVID-19 elements of UpSCALE. A second KAP survey is planned for November 2020 to measure change in KAP indicators and evaluate the role of adaptive learning practices alongside digital health tools, to enhance the capacity of APEs in this challenging COVID-19 pandemic.

## Discussion

Given the timing of the KAP survey at the start of the COVID-19 pandemic in Mozambique, during the first days of travel restrictions, and the fact that the survey was issued before training or supervision was provided, the response rate to the survey, at 28%, was in-line with other KAP studies conducted in similar settings and with similar populations in Uganda and Nepal [9, 11]. This was also the first time that users responded to a survey like this on the platform, hence the SMS reminders with instruction videos, which APEs reported finding useful to the implementation team. While it was important to recognise that already 15% of users relied on upSCALE for their COVID-19 information, local radio and health centres are clearly important sources of information and should continue to be utilised to communicate COVID-19 education messages. Only a quarter of the respondents were female, but since only 26% of the upSCALE app users are women, this proportion is to be expected.

Considering they had not yet received any formal COVID-19 training, there was a high level of knowledge of COVID-19 symptoms and transmission routes among the APEs. However, only 25% selected all correct symptoms and no others, when asked to list all. When asked to list all transmission routes, many people were able to list individual correct ways of transmitting the virus but only 16% listed the correct combination. A relatively high proportion (41%), lacked knowledge regarding the infectiousness of asymptomatic patients, this is currently being addressed through SMS messaging and the training modules. Subsequently, during September and October 2020, all APEs received one week of training on basic COVID-19 case detection and management from Ministry of Health, supported by UNICEF and other implementing partners.

Most of the respondents were able to list appropriate prevention measures. That being said, only 39% correctly listed the correct combination of recommended prevention methods. The vast majority said they are following government prevention guidelines at all times, but further education is required, e.g. confusion over the correct recommended handwashing length should be dispelled (only 15% correctly said 20 seconds). Misconceptions of those individuals most at risk and measures for those who develop COVID-19 symptoms are being clarified by the updated SMS message and digital modules that are delivered daily via upSCALE. Although many APEs could list some of the correct measures to take if symptoms occur, just 20% gave the correct answer when asked to list all. Awareness of risk for elderly and those with chronic diseases was high, however, when asked to list all high-risk groups, a very low proportion (less than two %) correctly listed the elderly, obese individuals and those with chronic illnesses as being high risk groups, and no other groups. A little less than half of the respondents (40%) also, incorrectly, included children as a high-risk group.

Although a high %age of survey participants were able to list some correct COVID-19 related symptoms, transmission routes and preventions methods, the proportion who answered each specific question correctly is considerably lower than other KAP surveys conducted among HCWs [8–11]. However, when reviewing the published data it is important to remember that knowledge of COVID-19 is constantly evolving and improving and KAP study results will depend largely on what point in a country's evolution of the pandemic they

occurred, making it hard to compare results. This KAP survey took place during the same time that Mozambique had imposed travel restrictions and opportunities for sharing information and increasing knowledge were severely restricted. Different studies use different cut off points to gauge what constitutes a 'good' understanding of COVID-19 and appropriate awareness of prevention and transmission. Furthermore, since this is the only KAP we are aware of focusing specifically on CHWs, the gap in available data about CHW COVID-19 KAP makes comparison among study results with those conducted in other countries or regions even more challenging. Additional research and information regarding KAP among CHWs are needed to be able to ensure they are getting the support necessary to effectively assist with their countries national COVID-19 response plans.

The SMS messaging content was based on WHO COVID-19 guidance, which is still evolving, therefore the development of these messages is an iterative process, and they are adapted and updated as guidance changes. The focus of the COVID-19 sensitive SMS messages have been adapted over time, based on the data flow of responses from the KAP survey, to correct any misconceptions. For example, SMS messages were sent out to reinforce the need for hand washing for a minimum of 20 seconds, to dispel myths of one minute, and reinforced messaging around the risk of transmission from asymptomatic populations.

A limitation of this KAP survey required the respondents to respond via the upSCALE application on their functioning mobile. This limited the sample size and response rates to the survey, particularly in Inhambane, where APES have older phones due to it being the first province where upSCALE was rolled out. This could also have led to a certain level of respondent bias, as typically approximately 20% of users have non-functioning phones and limited internet access, further biasing responses towards less remote areas. We have planned to also issue a shortened KAP survey using standard SMS messaging for the second round of the KAP survey planned for November 2020, to allow more users to respond.

## Conclusions

Adaptations to the upSCALE digital platform enabled the ministry of health supported by Malaria Consortium to rapidly gather essential insight on the COVID-19 knowledge gaps and misconceptions of APEs and then to help shape and target relevant messages, vital for ensuring a successful COVID-19 response. A daily series of SMS messages (currently totally more than 140,000), developed and shaped by knowledge gaps highlighted through the KAP study, were sent to a total of approximately 2,500 APEs (including all upSCALE app users) to reinforce MoH messaging and dispel misconceptions. The messages will continue to be sent. This study demonstrates how important it is to understand KAP of community health workers to COVID-19 to allow digital health tools, in this case the upSCALE app, to be adapted to better support CHWs as part of the COVID-19 pandemic response. The findings also show the continued need for further and ongoing education for these health workers in relation to COVID-19.

## Supporting information

**S1 Annex.**
(DOCX)

## Acknowledgments

The authors would like to acknowledge Charlotte Ward who contributed to the design of the questionnaire. We also express thanks to the APEs in Mozambique who kindly participated in

the study. The authors would like to thank the Ministry of Health for their support of the study.

## Author Contributions

**Conceptualization:** Vera Lacey Krylova, Laura Donovan, Joaquim Rebelo, Maria Rodrigues, Antonio Bulo, Carlos Ferraz, Humberto Rodrigues, Arantxa Roca-Feltrer, Kevin Baker.

**Data curation:** Vera Lacey Krylova, Poppy Farrow, Laura Donovan, Edson Zandamela, Antonio Bulo, Kevin Baker.

**Formal analysis:** Mitra Feldman, Edson Zandamela, Antonio Bulo, Kevin Baker.

**Funding acquisition:** Kevin Baker.

**Investigation:** Mitra Feldman, Kevin Baker.

**Methodology:** Vera Lacey Krylova, Laura Donovan, Edson Zandamela, Joaquim Rebelo, Humberto Rodrigues, Arantxa Roca-Feltrer, Kevin Baker.

**Project administration:** Joaquim Rebelo, Kevin Baker.

**Software:** Vera Lacey Krylova, Edson Zandamela, Antonio Bulo.

**Supervision:** Poppy Farrow, Maria Rodrigues, Humberto Rodrigues, Arantxa Roca-Feltrer, Kevin Baker.

**Validation:** Poppy Farrow, Edson Zandamela.

**Visualization:** Mitra Feldman, Vera Lacey Krylova, Edson Zandamela, Antonio Bulo.

**Writing – original draft:** Mitra Feldman, Kevin Baker.

**Writing – review & editing:** Mitra Feldman, Vera Lacey Krylova, Poppy Farrow, Laura Donovan, Edson Zandamela, Joaquim Rebelo, Maria Rodrigues, Antonio Bulo, Carlos Ferraz, Humberto Rodrigues, Arantxa Roca-Feltrer.

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
