## [Decision Letter · Decision Letter 0]

20 Nov 2020

PONE-D-20-32538

Community health worker knowledge, attitudes and practices towards COVID-19: learnings from an online cross-sectional survey using a digital health platform, UpSCALE, in Mozambique

PLOS ONE

Dear Dr. Kevin Nicholas Baker ,

Thank you for submitting your manuscript to PLOS ONE. After careful consideration, we feel that it has merit but does not fully meet PLOS ONE’s publication criteria as it currently stands. Therefore, we invite you to submit a revised version of the manuscript that addresses the points raised during the review process.

We look forward to receiving your revised manuscript.

Kind regards,

Francesco Di Gennaro

Academic Editor

PLOS ONE

Journal Requirements:

2.We note that you have indicated that data from this study are available upon request. PLOS only allows data to be available upon request if there are legal or ethical restrictions on sharing data publicly. For information on unacceptable data access restrictions, please see http://journals.plos.org/plosone/s/data-availability#loc-unacceptable-data-access-restrictions.

3.Thank you for stating the following in the Funding Section of your manuscript:

[upSCALE is funded by UK Aid (through UNICEF) and this work was funded through internal

Malaria Consortium funding.]

 [The funders had no role in study design, data collection and analysis, decision to publish, or preparation of the manuscript.]

We note that one or more of the authors are employed by a commercial company: Dimagi, Cape Town, South Africa

5. We note you have included a table to which you do not refer in the text of your manuscript. Please ensure that you refer to Table 1, 2 and 3 in your text; if accepted, production will need this reference to link the reader to the Table.

Additional Editor Comments (if provided):

Dear authors follow reviewers suggestion to improve your article

Reviewers' comments:

Reviewer's Responses to Questions

**Comments to the Author**

1. Is the manuscript technically sound, and do the data support the conclusions?

Reviewer #1: Yes

Reviewer #2: Partly

2. Has the statistical analysis been performed appropriately and rigorously? 

Reviewer #1: Yes

Reviewer #2: N/A

3. Have the authors made all data underlying the findings in their manuscript fully available?

Reviewer #1: Yes

Reviewer #2: Yes

4. Is the manuscript presented in an intelligible fashion and written in standard English?

Reviewer #1: Yes

Reviewer #2: No

5. Review Comments to the Author

Reviewer #1: The manuscript describe KAP of community health workers (Known as APEs in Mozambique) regarding Covid19 pandemic using the UpScale app. This is relevant and very timely as may improve the APEs conduct with regards to covid19.

A just have a few minor comments:

Methods section

Paragh 1: Please explain who Dimagi is (a private enterprise?)

Paragh 2: The authors state that "The SMS messages were accompanied by simple videos demonstrating how to install and complete the KAP survey on the application"... Could this be also a reason for the low response rate among the APEs? Is yes, please include in your discussion points.

Results

Paragh 1: There was a particular reason for the particular very low response rate in Inhambane province?

Discussion

Paragraphs 1 and 3: the authors indicate that this KAP survey took place during the same time that Mozambique was effectively in lock down. This is not factually correct as Mozambique never entered in a complete and effective lock down (level 4 restrictions). I would recommend to rephrase the statements.

Thanks

Reviewer #2: Short Title: I recommend the short title also include "..in Mozambique" at the end

Abstract: 1. Recommend to include the number of respondents and then indicate (28%) in parentheses. 2. Abbreviate percent as %

Manuscript

Technical approach: The study aims to evaluate the knowledge, attitudes and practices on COVID19 of community Health workers in 3 provinces of Mozambique using an adapted phone App.

1. Introduction- organize the literature review section by theme(Knowledge, attitudes and practices) so the text flow more clearly.

Study justification should be improved (?e.g.- to understand knowledge gaps among APEs and tailor interventions to identified gaps)

Describe what is the basic level of education of an APE

Did APE's receive any COVID training before or during the time data was being collected? This should be explained

2.Methods: Specify location of provinces mention- (North, South, central region of the country)

Clarify what is meant by " ...rapidly expanded the upSCALE platform". What does expansion mean?- the sentence needs to be rephrased and better explained what was done. when was this done? What was the main reason for expansion?

Enrollment or participants responded over a 2 month period (from June to August); During this time- did participants receive any training that could influence the APEs existing knowledge? what was the response rate per month (i.e how many respondents in June, July and in August?)

How did you gauge that the survey was "respondent friendly"?

the last paragraph of the methods section (page7) should be moved up earlier in the description of the program that was set up for the APEs.

3. Results: Low number of respondents in Inhambane province- are APE's evenly distributed across all three provinces? What was the response rate in Inhambane compared to the 2 other provinces? Were there provincial differences in how respondents answered the questions?

15% of respondents got covid information through the upscale APP: Were the SMS messages about COVID19 sent to ALL APEs? If so, why did only 15% report this as their source of information?

- Statistical analysis: has been performed appropriately and rigorously.

Recommendations for improvement by reformatting tables 2, 3 and 4 (include the number and % for each option). Were all questions answered? (i.e were there any gaps in responses to questions?)

Table 3: for question- on risk groups for COVID:-> recommend that authors include a row to indicate the number and % who gave all correct answers.

In terms of formatting for all sections of the tables- Recommend that the authors highlight the rows that show number and % of all correct answers given

PPE was reported in decreasing frequency from June to August: Please include the numbers and % for each month to enable readers understand the statement better. The question is whether there were fewer respondents in August compared to June. This is why it is important to describe how many respondents took part in each month.

The paragraph right below table 4 does not seem appropriate or relevant for the results section nor does it respond to the objectives of the study and recommend either removing it or including a clear description of the objectives of the study which would include assessing how the KAP findings were used.

Discussion: Were all APEs sent SMS messages? Why did only 15% say they learnt about covid through this method?

Clarify if there were any education or training sessions that took place for APEs on COVID before or during data collection.

The proportion of respondent who answered each specific question is reported to be lower than elsewhere: Please include the range seen in other places and compare with what was found in this study.

Recheck the grammar and spelling- in some places COVID is misspelled as COIVD; shorted instead of shortened etc.

Conclusion:

"The upscale App , can feasibly adapted to support CHWs in the COVID-19 pandemic" - this statement does not seem to be supported by the objectives /rationale of the study.

Feasibility as stated is also questionable since only 28% of expected participants responded.

The limitation of it being a self-administered questionnaire was not mentioned and could go against feasibility of using such approaches. This has not been adequately discussed on how or whether this can be dealt with.

4. Authors have stated that they made all data underlying the findings in their manuscript fully available

5. In general the manuscript is presented in an intelligible fashion and written in standard English- save for abbreviations, grammatical and spelling errors that should be corrected in several sections of the document; recommend an independent reader to make the corrections.

Ethical considerations: Reported to have been covered under an umbrella protocol. Participants gave consent. It was not clear whether the respondents are identifiable and how respondents were anonymized.

6. PLOS authors have the option to publish the peer review history of their article (what does this mean?). If published, this will include your full peer review and any attached files.

Reviewer #1: **Yes: **Pedro Aide

Reviewer #2: **Yes: **Charity Ndalama Alfredo, MBChB, MPH

---

## [Author Response · Author response to Decision Letter 0]

10 Dec 2020

Comments to the Author

1. Is the manuscript technically sound, and do the data support the conclusions?

Reviewer #1: Yes

Reviewer #2: Partly

2. Has the statistical analysis been performed appropriately and rigorously? 

Reviewer #1: Yes

Reviewer #2: N/A

3. Have the authors made all data underlying the findings in their manuscript fully available?

Reviewer #1: Yes

Reviewer #2: Yes

4. Is the manuscript presented in an intelligible fashion and written in standard English?

Reviewer #1: Yes

Reviewer #2: No

5. Review Comments to the Author

Reviewer #1: The manuscript describe KAP of community health workers (Known as APEs in Mozambique) regarding Covid19 pandemic using the UpScale app. This is relevant and very timely as may improve the APEs conduct with regards to covid19.

A just have a few minor comments:

Methods section

Paragh 1: Please explain who Dimagi is (a private enterprise?)

Author’s response: Thank you for seeking clarification on this. Dimagi is a private enterprise on whose platform, ComCare, the upSCALE applications have been developed. We have updated the text on page 5 as follows:

“The applications were developed on an open source platform designed specifically for use by frontline health workers (CommCare, Dimagi) “.

Paragh 2: The authors state that "The SMS messages were accompanied by simple videos demonstrating how to install and complete the KAP survey on the application"... Could this be also a reason for the low response rate among the APEs? Is yes, please include in your discussion points.

Author’s response: Thank you for seeking clarity of this. On consideration with the implementation team they felt the videos actually helped APEs to response – this was the first time they has responded to a survey on the platform and we have updated the text on page 15 to reflect this: 

“Given the timing of the KAP survey at the start of the COVID-19 pandemic in Mozambique, during the first days of travel restrictions, and the fact that the survey was issued without the ability to provide training or supervision, the response rate to the survey, at 28%, was in-line with other KAP studies conducted in similar settings and with similar populations in Uganda and Nepal (9, 11). This was also the first time that users responded to a survey like this on the platform, hence the SMS reminders with instruction videos, which APEs reported finding useful to the implementation team“.

Results

Paragh 1: There was a particular reason for the particular very low response rate in Inhambane province?

Author’s response: In discussion with the implementation team they highlighted that in Inhambane they have the oldest phones and don’t have solar panels to charge their phones. We have added text highlighted this as part of the limitations on page 17:

“A limitation of this KAP survey is that it required the respondents to respond via the upSCALE application on their functioning mobile. This limited the sample size and response rates to the survey, particularly in Inhambane, where APES have older phones due to it being the first province where upSCALE was rolled out “.

Discussion

Paragraphs 1 and 3: the authors indicate that this KAP survey took place during the same time that Mozambique was effectively in lock down. This is not factually correct as Mozambique never entered in a complete and effective lock down (level 4 restrictions). I would recommend to rephrase the statements.

Author’s response: Thank you for pointing this out and we have amended the text to reflect this as follows

“Given the timing of the KAP survey at the start of the COVID-19 pandemic in Mozambique, during the first days of travel restrictions……. This KAP survey took place during the same time that Mozambique had imposed travel restrictions and opportunities for sharing information and increasing knowledge were severely restricted“.

Reviewer #2: Short Title: I recommend the short title also include "...in Mozambique" at the end

Author’s response: Thank you – we have added this in the manuscript and we will attempt the same if space allows on the online system.

Abstract: 1. Recommend to include the number of respondents and then indicate (28%) in parentheses. 2. Abbreviate percent as %

Author’s response: Thank you for your suggestions and we have updated this in the tables.

Manuscript

Technical approach: The study aims to evaluate the knowledge, attitudes and practices on COVID19 of community Health workers in 3 provinces of Mozambique using an adapted phone App.

1. Introduction- organize the literature review section by theme (Knowledge, attitudes and practices) so the text flow more clearly.

Author’s response: Thank you for this suggestion and have reviewed the section as you suggested. We have organized this section by audience, as the focus of our paper is to point out the KAP of APEs in Mozambique. We hope this makes sense.

Study justification should be improved (?e.g.- to understand knowledge gaps among APEs and tailor interventions to identified gaps)

Author’s response: Thank you for your suggestion and we have decided, for clarity, to focus this manuscript on the KAP survey results solely and write another manuscript detailing the programmatic adaptions that followed. Therefore we have removed text detailing the SMS messages sent on page 7 and amended the study justification as follows on page 6:

“To better support tailoring and targeting of appropriate messages a COVID-19 KAP survey was developed and delivered to the APEs via upSCALE”.

Describe what the basic level of education of an APE is

Author’s response: Thank you and we have added an explanation of their education level, to highlight that it is basic, usually having completed 6th or 7th grade. We will also include that they receive 5 months initial training to be an APE and 1 month of practical sessions before commencing work as an APE. Their initial training on upSCALE is for 6 days. We have updated all of this on page 5 and 6:

“…agentes polivalentes elementares (APEs), are typically community members with a basic level of education, usually up to grade 6 or 7, who are trained to provide basic healthcare services and conduct health promotion activities in the remote areas in which they live. This includes one month’s training on the providing integrated community case management… APEs received 1 week of initial training on how to use upSCALE“.

Did APE's receive any COVID training before or during the time data was being collected? This should be explained

Author’s response: At the time we did the KAP survey APEs hadn’t received any training from the ministry. Subsequently, All APE received 1 week of training on basic COVID-19 case detection and management from Ministry of Health, supported by UNICEF and other implementing partners. We have updated this in the text as follows on page 15 & 16:

“Considering they had not yet received any formal COVID-19 training, there was a high level of knowledge of COVID-19 symptoms and transmission routes among the APEs ….. Subsequently, during September and October 2020, all APEs received one week of training on basic COVID-19 case detection and management from Ministry of Health, supported by UNICEF and other implementing partners “.

2.Methods: Specify location of provinces mention- (North, South, central region of the country)

Author’s response: Thank you, this has been updated in the text on page 6 as follows:

“…first in Inhambane in the south of Mozambique, followed by Cabo Delgado in the north, and then by Zambézia in central Mozambique…”.

Clarify what is meant by " ...rapidly expanded the upSCALE platform". What does expansion mean?- the sentence needs to be rephrased and better explained what was done. when was this done? What was the main reason for expansion?

Author’s response: Thank you for your suggestion on clarity of this point and we have updated the text as follows on page 6 – 

“…further developed the upSCALE platform and the use of telemedicine - through the use of short message service (SMS), training modules, monitoring of key indicators on routine services for women and children, stock control and disease surveillance”.

In a future manuscript we plan to detail the programmatic changes carried out in relation to COVID-19 on the upSCALE platform and the rationale for this, and therefore we have not included it in this manuscript in detail. 

Enrollment or participants responded over a 2 month period (from June to August); during this time- did participants receive any training that could influence the APEs existing knowledge? What was the response rate per month (i.e how many respondents in June, July and in August?)

Author’s response: Thank you for your suggestion – the APEs received training after the survey period in September and October and we have highlighted this as follows on page 16:

“Subsequently, during September and October 2020, all APEs received one week of training on basic COVID-19 case detection and management from Ministry of Health, supported by UNICEF and other implementing partners “.

How did you gauge that the survey was "respondent friendly"?

Author’s response: Thank you for your question and we did pre-test the survey with users before launch and have updated the manuscript with this detail on page 7 as follows:

“The draft survey was pretested with four users before launch to ensure it was accessible and respondent friendly“.

the last paragraph of the methods section (page7) should be moved up earlier in the description of the program that was set up for the APEs.

Author’s response: Thank you for your suggestion and as we are no longer including programmatic details of the broader upSCALE response to the pandemic we have removed the text from the manuscript and will include in a subsequent publication.

3. Results: Low number of respondents in Inhambane province- are APE's evenly distributed across all three provinces? What was the response rate in Inhambane compared to the 2 other provinces? Were there provincial differences in how respondents answered the questions?

Author’s response: Inhambane did have a lower response rate and we have added some explanation on this as part of the limitations on page 17:

“A limitation of this KAP survey is that it required the respondents to respond via the upSCALE application on their functioning mobile. This limited the sample size and response rates to the survey, particularly in Inhambane, where APES have older phones due to it being the first province where upSCALE was rolled out “.

15% of respondents got covid information through the upscale APP: Were the SMS messages about COVID19 sent to ALL APEs? If so, why did only 15% report this as their source of information?

Author’s response: Thank you for the question and yes, all users would have received SMS messages on their phone but only 15% remembered or realised this was linked to the application. This shows more work needs to be done to increase their awareness and we have updated the text to reflect this as follows on page 18

“The findings also show the continued need for further and ongoing education for these health workers in relation to COVID-19”.

- Statistical analysis: has been performed appropriately and rigorously.

Recommendations for improvement by reformatting tables 2, 3 and 4 (include the number and % for each option). Were all questions answered? (i.e were there any gaps in responses to questions?)

Author’s response: Thank you and we have updated the tables as suggested.

Table 3: for question- on risk groups for COVID:-> recommend that authors include a row to indicate the number and % who gave all correct answers.

In terms of formatting for all sections of the tables- Recommend that the authors highlight the rows that show number and % of all correct answers given

Author’s response: Thank you and we have updated the tables as suggested.

PPE was reported in decreasing frequency from June to August: Please include the numbers and % for each month to enable readers understand the statement better. The question is whether there were fewer respondents in August compared to June. This is why it is important to describe how many respondents took part in each month.

The paragraph right below table 4 does not seem appropriate or relevant for the results section nor does it respond to the objectives of the study and recommend either removing it or including a clear description of the objectives of the study which would include assessing how the KAP findings were used.

Author’s response: Thank you and we have updated the tables as suggested. We have also removed the paragraph relating to programmatic elements of upSCALE and will include that in the subsequent manuscript. 

Discussion: Were all APEs sent SMS messages? Why did only 15% say they learnt about covid through this method?

Clarify if there were any education or training sessions that took place for APEs on COVID before or during data collection.

The proportion of respondent who answered each specific question is reported to be lower than elsewhere: Please include the range seen in other places and compare with what was found in this study.

Recheck the grammar and spelling- in some places COVID is misspelled as COIVD; shorted instead of shortened etc.

Author’s response: Thank you for these questions and apologies for selling errors these have now been updated. We did address the lower response rate in the text as follows on page 15:

“Given the timing of the KAP survey at the start of the COVID-19 pandemic in Mozambique, during the first days of lockdown travel restrictions, and the fact that the survey was issued before training or supervision was provided, the response rate to the survey, at 28 %, was in-line with other KAP studies conducted in similar settings and with similar populations in Uganda and Nepal”.

Conclusion:

"The upscale App can feasibly adapted to support CHWs in the COVID-19 pandemic" - this statement does not seem to be supported by the objectives /rationale of the study.

Feasibility as stated is also questionable since only 28% of expected participants responded.

The limitation of it being a self-administered questionnaire was not mentioned and could go against feasibility of using such approaches. This has not been adequately discussed on how or whether this can be dealt with.

Author’s response: Thank you and we have updated the conclusion as follows:

"This study demonstrates how important it is to understand the KAP of community health workers to COVID-19 to allow digital health tools, in this case the upSCALE app, to be adapted to better support CHWs as part of the COVID-19 pandemic response".

4. Authors have stated that they made all data underlying the findings in their manuscript fully available

5. In general the manuscript is presented in an intelligible fashion and written in standard English- save for abbreviations, grammatical and spelling errors that should be corrected in several sections of the document; recommend an independent reader to make the corrections.

Author’s response: Thank you and we have checked and edited the text as suggested. 

Ethical considerations: Reported to have been covered under an umbrella protocol. Participants gave consent. It was not clear whether the respondents are identifiable and how respondents were anonymized.

Author’s response: Thank you and we have updated the ethics text as follows on page 7:

“The questionnaire contained a consent section that included a statement about its purpose, objectives, voluntary participation, and a declaration of confidentiality and anonymity. All responses were submitted anonymously via the UpSCALE platform”.

We hope this addresses all points raised and thank you for your ongoing consideration

---

## [Decision Letter · Decision Letter 1]

21 Dec 2020

Community health worker knowledge, attitudes and practices towards COVID-19: learnings from an online cross-sectional survey using a digital health platform, UpSCALE, in Mozambique

PONE-D-20-32538R1

Dear Dr. Baker,

We’re pleased to inform you that your manuscript has been judged scientifically suitable for publication and will be formally accepted for publication once it meets all outstanding technical requirements.

Kind regards,

Francesco Di Gennaro

Academic Editor

PLOS ONE

Additional Editor Comments (optional):

dear authors congratulations

Reviewers' comments:

Reviewer's Responses to Questions

**Comments to the Author**

1. If the authors have adequately addressed your comments raised in a previous round of review and you feel that this manuscript is now acceptable for publication, you may indicate that here to bypass the “Comments to the Author” section, enter your conflict of interest statement in the “Confidential to Editor” section, and submit your "Accept" recommendation.

Reviewer #1: All comments have been addressed

Reviewer #2: All comments have been addressed

2. Is the manuscript technically sound, and do the data support the conclusions?

Reviewer #1: Yes

Reviewer #2: (No Response)

3. Has the statistical analysis been performed appropriately and rigorously? 

Reviewer #1: Yes

Reviewer #2: (No Response)

4. Have the authors made all data underlying the findings in their manuscript fully available?

Reviewer #1: Yes

Reviewer #2: (No Response)

5. Is the manuscript presented in an intelligible fashion and written in standard English?

Reviewer #1: Yes

Reviewer #2: (No Response)

6. Review Comments to the Author

Reviewer #1: The authors have provided clear responses and the manuscript has improved with the suggested reviews. I have no further comments.

Reviewer #2: (No Response)

7. PLOS authors have the option to publish the peer review history of their article (what does this mean?). If published, this will include your full peer review and any attached files.

Reviewer #1: **Yes: **Pedro Aide

Reviewer #2: **Yes: **Charity Ndalama Alfredo MBChC, MPH

---

## [Editor Report · Acceptance letter]

1 Feb 2021

PONE-D-20-32538R1 

Community health worker knowledge, attitudes and practices towards COVID-19: learnings from an online cross-sectional survey using a digital health platform, UpSCALE, in Mozambique 

Dear Dr. Baker:

I'm pleased to inform you that your manuscript has been deemed suitable for publication in PLOS ONE. Congratulations! Your manuscript is now with our production department. 

Kind regards, 

on behalf of

Dr. Francesco Di Gennaro 

Academic Editor

PLOS ONE